# Clinical Outcome of Rotational Atherectomy in Calcified Lesions in Korea-ROCK Registry

**DOI:** 10.3390/medicina57070694

**Published:** 2021-07-07

**Authors:** Kyusup Lee, Ji-hoon Jung, Myunhee Lee, Dae-Won Kim, Mahn-Won Park, Ik-Jun Choi, Jae-Hwan Lee, Jang-Hoon Lee, Sang-Rok Lee, Pil-Hyung Lee, Seung-Whan Lee, Ki-Dong Yoo, Kyeong-Ho Yun, Hyun-Jong Lee, Sung-Ho Her

**Affiliations:** 1Department of Cardiology, Daejeon St. Mary’s Hospital, College of Medicine, The Catholic University of Korea, Seoul 06591, Korea; ajobijh@hanmail.net (K.L.); myunheelee@catholic.ac.kr (M.L.); mirinesilver@catholic.ac.kr (D.-W.K.); pmw6193@catholic.ac.kr (M.-W.P.); 2Korea Institute of Toxicology, Daejeon 34114, Korea; jihoon.jung@kitox.re.kr; 3Department of Cardiology, Incheon St. Mary’s Hospital, College of Medicine, The Catholic University of Korea, Incheon 21431, Korea; mrfasthand@catholic.ac.kr; 4Department of Cardiology in Internal Medicine, Chungnam National University Hospital, Daejeon 35015, Korea; myheart@cnuh.co.kr; 5Department of Internal Medicine, Kyungpook National University Hospital, Daegu 41944, Korea; ljhmh75@knu.ac.kr; 6Department of Cardiology, Chonbuk National University Hospital, Jeonju 54907, Korea; medorche@naver.com; 7Asan Medical Center, Department of Cardiology, University of Ulsan College of Medicine, Seoul 05505, Korea; pilmo11@hanmail.net (P.-H.L.); seungwlee@amc.seoul.kr (S.-W.L.); 8Department of Cardiology, St. Vincent’s Hospital, College of Medicine, The Catholic University of Korea, Seoul 06591, Korea; yookd@catholic.ac.kr; 9Regional Cardiocerebrovascular Center, Department of Cardiovascular Medicine, Wonkwang University Hospital, Iksan 54538, Korea; ards7210@wonkwang.ac.kr; 10Department of Internal Medicine, Sejong General Hospital, Bucheon 14754, Korea; untouchables00@hanmil.net

**Keywords:** percutaneous coronary intervention, rotational atherectomy, drug-eluting stent, clinical outcome

## Abstract

*Background and Objectives*: Data is still limited regarding clinical outcomes of rotational atherectomy (RA) after percutaneous coronary intervention. We sought to evaluate clinical outcomes of RA. *Materials and Methods:* This multi-center registry enrolled patients who underwent RA during PCI from nine tertiary centers in Korea between January 2010 and October 2019. The primary endpoint was target-vessel failure (TVF; the composite outcome of cardiac death, target-vessel spontaneous myocardial infarction, or target-vessel revascularization). *Results*: Of 540 patients (583 lesions), the mean patient age was 71.4 ± 0.4 years, 323 patients (59.8%) were men, and 305 patients (56.5%) had diabetes mellitus. Technical success rate was 96.4%. In-hospital major adverse cerebral and cardiac events occurred in 63 cases (10.8%). At 1.5 years, 72 (16.0%) of TVFs were occurred. We evaluated independent predictors of TVF, which included current smoker (hazard ratio (HR), 1.92; 95% confidence interval (CI), 1.17–3.16; *p* = 0.01), chronic renal disease (HR, 1.87; 95% CI, 1.14–3.08; *p* = 0.013), history of cerebrovascular attack (HR, 2.14; 95% CI, 1.24-3.68; *p* = 0.006), left ventricle ejection fraction (HR, 0.98; 95% CI, 0.97–0.999; *p* = 0.037), and left main disease (HR, 1.94; 95% CI, 1.11–3.37; *p* = 0.019). *Conclusions:* From this registry, we demonstrated acceptable success rates, in-hospital and mid-term clinical outcomes of RA in the DES era.

## 1. Introduction

With the development of device technology, there have been expanded indications of percutaneous coronary intervention (PCI) to more challenging cases. Moreover, as the elderly population increases, we can get a more chance to meet patients with complex calcified coronary lesions in clinical practice [1].

With this current change, methods for modifying calcified coronary lesions have been more and more important (such as a non-compliant balloon, cutting/scoring balloon, orbital atherectomy, intravascular lithotripsy, and rotational atherectomy (RA)) [2,3,4,5]. The RA system is composed of a high-speed rotating diamond-coated burr that modifies calcified lesions. Therefore, it is a useful device for plaque modification of those lesions to facilitate balloon and stent delivery [6,7]. However, in our daily practice, patients qualified for RA are often “non-option” patients due to their comorbidity. Operators should be concerned about procedure-related complications and adverse clinical outcomes [8,9,10,11,12]. With the widespread use of second-generation drug-eluting stents (DES), optimal medical treatment, including statin and intravascular image, there has been a need for updated clinical results of RA in real-world practice. Therefore, we designed this study to introduce our registry and to investigate procedural outcomes, in-hospital events, and clinical outcomes of RA in the DES era.

## 2. Materials and Methods

### 2.1. Study Design and Population

The study population consisted of 540 patients (583 lesions) with calcified coronary artery disease (CAD) who underwent PCI using RA between January 2010 and October 2019 at 9 tertiary centers, from the ‘ROtational atherectomy in Calcified lesions in Korea (ROCK)’ registry. Data were collected at each site using a standardized case report form to record demographic and clinical characteristics and procedural and follow-up data. Follow-up data were collected retrospectively, based on medical records and a physician or patient interview. The local ethics committee approved this study of each hospital.

Between 2010 and 2019, consecutive patients with heavily calcific coronary lesions and significant stenosis (stenosis ≥ 70% of vessel diameter) who underwent PCI using RA were retrospectively enrolled from each institutional database. After reviewing angiography, two lesions could not be performed RA were dropped out from the registry. One experienced coronary perforation and cardiac tamponade before the RA procedure. The other could not pass the wire through the target lesion. Chronic kidney disease (CKD) was defined as an estimated glomerular filtration rate <60 mL/min/1.73 m^2^, as calculated using the Modification of Renal Diet equation from baseline serum creatinine [13].

### 2.2. Procedure

The treatment strategy, including decisions of performing/timing of RA, burr size, and selection of vascular access, was at the discretion of the attending cardiologists with careful consideration of clinical risk factors, anatomical complexity, patients’ conditions. Standard techniques and management guided all procedures. All RA procedures were performed using the Rotablator^TM^ RA system (Boston Scientific, Marlborough, MA, USA). During RA, pauses in ablation runs and intracoronary nitroglycerin and/or verapamil were used to avoid coronary spasm and slow flow phenomenon. Antiplatelet therapy and peri-procedural anticoagulation were performed following the accepted guidelines [14,15].

### 2.3. Study Outcomes

The primary endpoint of the study was the occurrence of target-vessel failure (TVF), defined as cardiac death, target-vessel spontaneous myocardial infarction (MI), or target vessel revascularization (TVR). The secondary endpoint was all-cause death, cardiac death, any MI, target-vessel spontaneous MI, any repeat revascularization (RR), TVR, target-lesion revascularization (TLR), stent thrombosis (ST), coronary artery bypass grafting (CABG) surgery, cerebrovascular accident (CVA), and bleeding.

Technical success was defined as the achievement of residual stenosis < 30% in the presence of grade III Thrombolysis in myocardial infarction flow. Procedural success was defined as achieving technical success without in-hospital major adverse cerebral and cardiac events (MACCEs), including in-hospital death, in-hospital CVA, urgent revascularization (CABG or PCI), peri-procedural MI or ST during the index hospitalization period. Procedural complications included cardiac tamponade, coronary perforation, severe coronary dissection, defined from The National Heart, Lung, and Blood Institute classification system as type D, E, and F, temporary pacemaker insertion, contrast-induced nephropathy, or in-hospital bleeding.

We investigated procedure time, radiation dose, and contrast amount to assess procedural efficiency and safety. We also reviewed contrast-induced nephropathy (CIN) after PCI, which was defined as the impairment of kidney function-measured as either a 25% increase in serum creatinine from baseline or a 0.5 mg/dL increase in absolute serum creatinine value within 48–72 h after the procedure.

Death was defined as death from any cause. Target-vessel spontaneous MI was spontaneous MI clearly attributable to the target vessel. Spontaneous MI was defined as any creatine kinase-myocardial band or troponin increase above the upper limit of the normal range with ischemic symptoms or signs during follow-up after discharge. Peri-procedural MI was defined as peak elevations of the creatine kinase-myocardial band of >10-fold above the upper reference limit within 48 h after the procedure [16]. RR was defined as any percutaneous or surgical revascularization in any vessel. TVR was defined as any percutaneous or surgical revascularization of the treated vessel. TLR was defined as any percutaneous or surgical revascularization of the treated lesion. CVA was defined as a focal neurological deficit of central origin lasting >24 h, confirmed by a neurologist and imaging. All clinical events were confirmed by source documentation collected at each hospital and centrally adjudicated by an independent group of clinicians unaware of the revascularization type.

### 2.4. Statistical Analyses

Continuous variables were presented as mean ± standard deviation (SD) and compared using the Student *t*-test or Mann–Whitney *U* test. Categorical variables were presented as counts (percentages) and compared using the chi-square or Fisher exact test, as appropriate. Event rates were estimated on Kaplan–Meier estimates in time-to-first-event analyses, and they were compared using the log-rank test. A univariate Cox regression analysis was performed to obtain the hazard ratio (HR) for clinical outcomes. Then, to find out the independent predictors with the clinical outcomes, a multivariate Cox proportional hazard regression model was performed using important clinical covariates including clinically relevant variables and statistically significant variables with a *p*-value < 0.1 by univariate analysis. All reported p values were two-sided, and *p* values < 0.05 were considered statistically significant. Statistical analysis was performed with Statistical Package for Social Sciences, version 18.0.0 (SPSS Inc., Chicago, IL, USA).

## 3. Results

### 3.1. Baseline Patient, Lesion, and Procedural Characteristics

From January 2010 to October 2019, a total of 540 patients (583 lesions) who received PCI using RA were enrolled (Figure 1). Baseline patient characteristics are summarized in Table 1. The mean age was 71.4 ± 0.4 years (range, 19–104 years), and 323 (59.8%) patients were male gender. Three hundred five (56.5%) patients had diabetes mellitus. More than 60% of patients were diagnosed with the acute coronary syndrome. The mean left ventricular ejection fraction was 53.0 ± 13.4. Dual antiplatelet therapy (DAPT) and statin were prescribed well at discharge (95.9% and 92.4%, respectively).

Values are presented as n (%) or mean ± SD. Abbreviations: SD, standard deviation; IQR, interquartile range; BMI, body mass index; CAD, coronary artery disease; NSTEMI, non-ST elevation myocardial infarction; STEMI, ST-elevation myocardial infarction; ACS, acute coronary syndrome; MI, myocardial infarction; PCI, percutaneous coronary intervention; CABG, coronary artery bypass grafting; LV EF, left ventricle ejection fraction; DAPT, dual antiplatelet therapy; ACEI/ARB, angiotensin-converting-enzyme inhibitor/angiotensin II receptor blocker; NOAC, non-vitamin K oral anticoagulant.

The majority of lesions were classified as type B2 (10.1%) or C (82.3%) according to the American College of Cardiology and the American Heart Association (ACC/AHA) classification. Left anterior descending (LAD), including left main (LM), left circumflex (LCX) coronary artery, and right coronary artery (RCA) was involved in 65.9%, 10.1%, and 24.0%, respectively. LM and multivessel disease included in 14.1% and 80.1%, respectively (Table 2).

Values are presented as n (%) or mean ± SD. Abbreviations: SD, standard deviation; ACC/AHA, American College of Cardiology/American Heart Association; LAD, left anterior descending; LCX, left circumflex; RCA, right coronary artery; LM, left main; POBA, plain old balloon angioplasty; DEB, drug-eluting balloon; IVUS, intravascular ultrasound; OCT, optical coherence tomography.

Detailed procedural characteristics were described in Table 2. The femoral approach was used in 55.4% of cases, whereas other cases used the radial approach. Stent implantation was performed in 548 (94.0%) patients, plain old balloon angioplasty (POBA) in six (1.0%), and drug-eluting balloon (DEB) in 16 (2.7%). The others (11 [1.9%] patients) could not perform anything due to technical failure. Intravascular ultrasound (IVUS) or optical coherence tomography were used in 46.0% of cases. The most frequently used burr size (maximum) was 1.5 mm (52.9%), followed by 1.25 mm, 1.75 mm, 2.0 mm, and 2.5 mm (26.9%, 18.4%, 1.7%, and 0.2%).

The number of stents per patient and per lesion was 2.38 ± 1.18 and 1.66 ± 0.68. Mean stent diameter, total stent length, and stent length per lesion were 2.99 ± 0.38, 69.02 ± 35.66, and 49.48 ± 21.13, respectively.

### 3.2. Procedural Outcomes

The rate of technical and procedural success was 96.4% and 86.1%. Procedure time was 63.4 (45.0–97.0) minutes. In addition, the amount of contrast used was 210 mL (interquartile range [IQR)]: 150–300 mL), and the radiation dose was 2685 mGy (IQR: 1512–4800 mGy) (Table 3).

As shown in Table 3, in-hospital major adverse cerebral and cardiac events (MACCEs) occurred in 62 cases (10.6%), mainly driven by peri-procedural myocardial infarction (46 cases, 7.9%). In-hospital death and in-hospital CVA occurred in 11 (1.9%) and 2 (0.3%) patients. Severe coronary dissection (Type D, E, and F) and coronary perforation occurred in 35 (6.0%) and 11 (1.9%) cases. Cardiac tamponade requiring intervention occurred in 3 cases (0.5%). Urgent revascularization was needed in nine patients (1.5%; 2 CABG; 7 PCI).

### 3.3. Clinical Outcomes

Median follow-up duration was 1.5 (IQR, 0.7–2.9) years. At 1.5 years, 72 TVFs (16.0%) occurred and all-cause death occurred in 37 (8.4%), cardiac death in 33 (6.9%), spontaneous MI in 18 (4.2%), target-vessel spontaneous MI in 9 (2.1%), any RR in 53 (12.8%), TVR in 41 (9.8%), TLR in 34 (8.2%), CVA in 10 (2.0%), and ST in 6 (1.2%) (Figure 2). One patient underwent urgent CABG at 11 days after the index procedure.

We evaluated independent predictors of TVF, which included current smoker (hazard ratio [HR], 1.92; 95% confidence interval (CI), 1.17–3.16; *p* = 0.01), chronic renal disease (HR, 1.87; 95% CI, 1.14–3.08; *p* = 0.013), history of cerebrovascular attack (HR, 2.14; 95% CI, 1.24–3.68; *p* = 0.006), left ventricle ejection fraction (HR, 0.98; 95% CI, 0.97–0.999; *p* = 0.037), and left main disease (HR, 1.94; 95% CI, 1.11–3.37; *p* = 0.019) (Table 4).

### 3.4. Subgroup Analysis among Patients that Received PCI Successfully According to Clinical Presentation

To demonstrate the high mortality rate of our registry (27.1% during the entire follow-up period, 0–10.1 years), we performed a subgroup analysis of patients who received PCI successfully (n = 541). We investigated the long-term clinical outcomes according to clinical presentation (ACS versus stable angina). Kaplan–Meier curves showed poorer clinical outcomes regarding the TVF and mortality in ACS group compared to stable angina group (21.7% vs. 14.5%, log-rank *p* = 0.005; 14.1% vs. 6.8%, log-rank *p* < 0.001, respectively) (Appendix A). Multivariable cox regression analysis revealed that ACS was an independent predictor of mortality (HR, 1.94; 95% CI, 1.002–3.75; *p* = 0.049) (Appendix A).

## 4. Discussion

Major findings from our study were as follows: (1) Technical and procedural success rate of RA was acceptable in heavily calcified and diffuse coronary lesions; (2) RA during PCI was safe and efficient for revascularization regarding in-hospital MACCEs and procedural complications, as well as procedural outcomes (procedure time, contrast amount, or radiation dose); (3) the independent predictors of TVF were identified from our registry.

This ‘ROtational atherectomy in Calcified lesions in Korea (ROCK)’ registry is the largest, all-comer, multi-center registry, including patients undergoing RA in Korea. Patients who were received PCI were implanted with drug-eluting stents (DESs), especially 2nd generation DES, except for one patient with 1st generation DES, which might reflect the clinical outcomes of RA in the 2nd-generation DES era. As described previously, DAPT and statin were prescribed above 90% of the study population, which would reflect that these patients were treated as current guidelines [14,17,18].

All cases except for a few chronic total occlusion or in-stent restenosis cases were calcified and diffuse disease, represented by 82.4% of lesions classified as type C (ACC/AHA classification). In 38% of cases, RA was done due to unsuccessful 2.0 mm sized or bigger balloon predilatation. In the others, RA was done promptly based on the operator's decision. It was based on angiographic findings, the calcification arc revealed by intravascular images, the friction of smaller balloons or wires, etc. Although, we provided data of acceptable efficacy outcomes regarding success rate and procedural outcomes. We achieved final TIMI 3 in 96.4% of all cases, similar to the previous studies [1,12,19,20,21]. The absolute values of procedure time and contrast amount were smaller compared to the previous study [20].

The study population consisted of patients with advanced coronary atherosclerosis and high comorbidity. Indeed, diabetes mellitus accounted for more than half of the study population, and more than 80% were multivessel diseases. From the current guideline, for revascularization of multivessel disease accompanied by diabetes, we should consider CABG [22,23]. However, because of high-risk peri-operative mortality and morbidity and patients’ preference, operators chose the PCI with RA instead of CABG for those patients in this registry. Although, in-hospital MACCEs and procedural complications showed feasible and acceptable results compared to other previous studies [1,12,19,20,21]. However, in-hospital MACCEs were developed in 10.6% of our registry, mainly driven by peri-procedural MI, which showed higher events than previous studies. We thought plausible explanations as follows; (1) because we included more than 60% of ACS patients; (2) events were recorded according to 10 times elevation of the cardiac enzyme (CK-MB) with medical records, would be over-estimated. Coronary perforation occurred in 1.9% of cases, consistent with previous studies (0.5–2.3%) [12,19,20,21].

As shown in Table 5, our study showed relatively high mortality similar to that of the J2T Multicenter registry [1]. We found one possible explanation from baseline clinical demographics in both of the two studies, which might be due to a large proportion (more than 50%) of the acute coronary syndrome (ACS). Our subgroup analysis demonstrated ACS was an independent predictor of mortality among patients who received PCI successfully.

Patients received 2nd-generation DES were included in 36.3% (J2T Multicenter registry) [1] and 69.3% (ROTATE registry) [20]. In our registry, as mentioned above, we included 99.8% of 2nd-generation DES among patients treated with stent implantation. This homogeneity of implanted stents is the unique and valuable strength of our registry. Moreover, nearly half of the cases were performed procedures with intravascular image guidance. We could evaluate the benefit of intravascular guidance in RA with our data.

TVF occurred in 16.0% at 1.5 years, which showed acceptable clinical outcomes to previous studies considering that our study population consisted of more elderly patients, ACS patients, including ST-elevation MI, and the long-term follow-up period. We summarized clinical outcomes compared with previous studies of RA (Table 5).

### Limitation

Our study has several limitations. This was a non-randomized, observational, single-arm, retrospective study. Due to the relatively large number of centers included in the study over the long study period, there may be confounders that have not been accounted for. Second, the registry included a relatively limited number of patients and insufficient follow-up duration to evaluate the long-term clinical outcomes. Third, parameters of lesion complexity (such as SYNTAX score or quantitative coronary angiography) were not available. Therefore, our findings should be further evaluated after reassessing those parameters.

## 5. Conclusions

Our study has provided a large registry for in-hospital and long-term clinical outcomes of RA in the 2nd-generation DES era, which showed acceptable success rates, in-hospital, and long-term clinical outcomes.

## Figures and Tables

**Figure 1 medicina-57-00694-f001:**
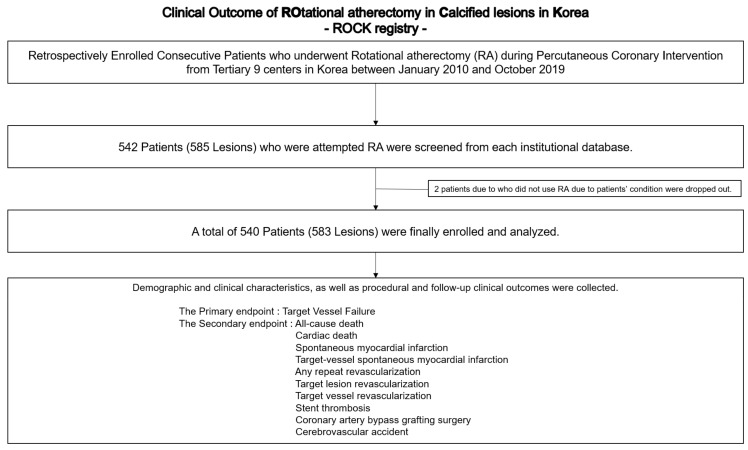
Study Population Flow Chart.

**Figure 2 medicina-57-00694-f002:**
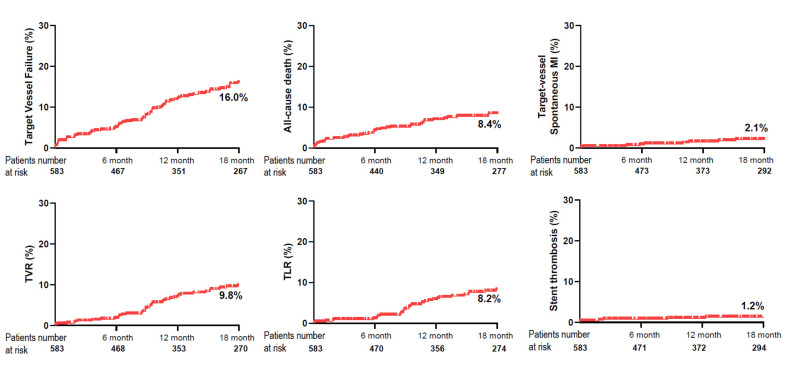
Kaplan-Meier Curves for Clinical Outcomes at 18 months.

**Table 1 medicina-57-00694-t001:** Baseline characteristics of the patients.

Characteristics	Mean ± SD, n (%) or Median (IQR) (n = 540)
Age (years)	71.4 ± 0.4
Male	323 (59.8)
BMI (kg/m^2^)	24.2 ± 3.9
Diabetes mellitus	305 (56.5)
Insulin use	72 (13.3)
Hypertension	415 (76.9)
Hyperlipidemia	235 (43.5)
Current smoker	103 (19.1)
Clinical presentation	
Silent ischemia	37 (6.9)
Stable angina	176 (32.6)
Unstable angina	175 (32.4)
NSTEMI	133 (24.6)
STEMI	19 (3.5)
Presented as ACS	327 (60.6)
Chronic kidney disease	96 (17.8)
Prior MI	66 (12.2)
Prior PCI	138 (25.6)
Prior CABG	24 (4.4)
Peripheral vascular disease	39 (7.2)
History of heart failure	77 (14.3)
Atrial fibrillation	49 (9.1)
LV EF, %	53.0 ± 13.4
Medication at discharge	
Aspirin	529 (98.0)
P2Y12 Inhibitor	530 (98.1)
DAPT	518 (95.9)
B-blocker	381 (70.6)
ACEI/ARB	341 (63.1)
Statin	499 (92.4)
NOAC	17 (2.9)

**Table 2 medicina-57-00694-t002:** Lesion and procedural characteristics.

Characteristics	Mean ± SD, n (%) or Median (IQR) (n = 583)
ACC/AHA classification	
Type B2/C	59 (10.1)/480 (82.4)
Target vessel	
LAD	384 (65.9)
LCX	59 (10.1)
RCA	140 (24.0)
LM disease	82 (14.1)
Multivessel disease	467 (80.1)
Femoral approach	323 (55.4)
Treatment type	
POBA	6 (1.0)
DEB	17 (2.9)
Stent implantation	548 (94.0)
Failure	12 (2.0)
IVUS or OCT	268 (46.0)
Escalation of Burr	91 (15.6)
Maximum Burr size, mm	
1.25	157 (26.9)
1.5	308 (52.9)
1.75	107 (18.4)
2.0	10 (1.7)
2.5	1 (0.2)
Access size	
5 F	5 (0.9)
6 F	215 (36.9)
7 F	326 (55.9)
8 F	37 (6.3)
Total number of stents	2.38 ± 1.18
Number of stents (target-vessel)	1.66 ± 0.68
Mean stent diameter, mm (target-vessel)	2.99 ± 0.38
Total stent length, mm	69.02 ± 35.66
Stent length, mm (target-vessel)	49.48 ± 21.13

**Table 3 medicina-57-00694-t003:** In-hospital Events and Procedural Outcomes.

	Mean ± SD, n (%) or Median (IQR)
Technical success	562 (96.4)
Procedural success	508 (87.1)
Contrast amount, mL	210 (150–300)
Procedure time, min	63.4 (45.0–97.0)
Radiation dose, mGy	2685 (1512–4800)
In-hospital MACCEs	62 (10.6)
In-hospital death	11 (1.9)
Procedure-related myocardial infarction	46 (7.9)
Urgent revascularization	9 (1.5)
In-hospital stroke	2 (0.3)
Procedural Complications	
Cardiac tamponade requiring intervention	3 (0.5)
Coronary perforation	11 (1.9)
Severe coronary dissection immediate after RA procedure *	35 (6.0)
Temporary pacemaker during RA procedure	19 (3.3)
Contrast-Induced Nephropathy	19 (3.3)
In-hospital bleeding	27 (4.6)

***** Type D, E, and F, defined from The National Heart, Lung, and Blood Institute (NHLBI) classification system. MACCE, major adverse cardiac and cerebral event; RA, rotational atherectomy; SD, standard deviation; IQR, interquartile range.

**Table 4 medicina-57-00694-t004:** Univariable and multivariable cox regression analysis of independent predictors of target-vessel failure.

	Univariate		Multivariate *	
Variables	HR (95% CI)	*p*	HR (95% CI)	*p*
Age	0.99 (0.97–1.01)	0.43		
Male gender	1.12 (0.74–1.70)	0.59		
Current smoker	1.67 (1.06–2.65)	0.028	1.92 (1.17–3.16)	0.01
HTN	0.97 (0.60–1.57)	0.91		
Diabetes mellitus	1.39 (0.91–2.11)	0.13		
Hyperlipidemia	0.59 (0.38–0.92)	0.018		
Chronic kidney disease	2.12 (1.34–3.35)	0.001	1.87 (1.14–3.08)	0.013
Family history of CAD	1.50 (0.37–6.11)	0.57		
Prior MI	0.60 (0.28–1.29)	0.19		
Prior PCI	1.10 (0.69–1.73)	0.70		
Prior CABG	0.91 (0.33–2.48)	0.85		
History of cerebrovascular attack	2.15 (1.32–3.50)	0.002	2.14 (1.24–3.68)	0.006
Peripheral artery disease	1.69 (0.88–3.25)	0.12		
Atrial fibrillation	1.54 (0.82–2.90)	0.18		
LV EF (%)	0.98 (0.96–0.99)	0.003	0.98 (0.97–0.999)	0.037
Presented as ACS	1.61 (1.03–2.51)	0.035	1.54 (0.95–2.51)	0.08
MVD	0.95 (0.56–1.61)	0.86		
LM disease	1.68 (1.02–2.75)	0.041	1.94 (1.11–3.37)	0.019
Mean stent diameter per target vessel	0.97 (0.55–1.71)	0.92		
Total number of stents	1.23 (1.04–1.44)	0.014	1.19 (0.996–1.41)	0.06
Total stent length	1.007 (1.001–1.012)	0.026		

* adjusted by age, gender, current smoker, diabetes mellitus, chronic kidney disease, history of cerebrovascular attack, LV EF, presented as an acute coronary syndrome, left main disease, the total number of stents, and total stent length HR, hazard ratio; CI, confidence interval; HTN, hypertension; CAD, coronary artery disease; MI, myocardial infarction; PCI, percutaneous coronary intervention; CABG, coronary artery bypass grafting surgery; LV EF, left ventricle ejection fraction; ACS, acute coronary syndrome; MVD, multivessel disease; LM, left-main.

**Table 5 medicina-57-00694-t005:** Comparison with previous studies of rotational atherectomy including technical success and follow-up clinical outcomes.

	All Study Population	Era	Technical Success, %	Follow-Up Duration	All-Cause Death, %	TVR, %	TLR, %	ST, %	CVA, %
Okai et al. [1]	1090	DES	96.2	3.8 (IQR 1.9–6.1) years	24.2	21.4	16.2	1.3	4.7
Kawamoto et al. [20]	985	8.8% BMS91.2% DES	99.1	2 years	9.5	19.8	16.6	1.8	1.0
Cortese et al. [21]	1397	17% POBA40.7% BMS42% DES	99.3	2.4 (0.8–5.7) years	6.4	-	11.7	1.4	-
Rathore et al. [19]	516	24.2% BMS75.8% DES	99.4	6–9 months	-	-	14.4	0.8 *	-
Abdel-Wahab et al. [12]	205	DES	98.0	1.3 (0.08–7) years	9.0	11.2	7.9	1.0	-
Our study **	540	DES	96.4	1.5 (IQR 0.7–2.9) years	8.4	9.8	8.2	1.2	2.0

* Includes only late stent thrombosis. ** Event rates at 18 months were estimated on Kaplan–Meier estimates in time-to-first-event analyses. TVR, target-vessel revascularization; TLR, target-lesion revascularization; ST, stent thrombosis; CVA, cerebrovascular accident; DES, drug-eluting stent; IQR, interquartile range; BMS, bare metal stent; POBA, plain old balloon angioplasty.

## Data Availability

The data presented in this study are available on request from the corresponding author.

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
