# Peer review of "Clinical Outcome of Rotational Atherectomy in Calcified Lesions in Korea-ROCK Registry"

_medicina, 2021, doi:10.3390/medicina57070694_

Round 1
Reviewer 1 Report
Introduction :
I would suggest adding a short comment regarding other methods of calcified lesion modification (NC balloon; cutting/scoring balloon; orbital atherectomy; Shockwave lithotripsy). Also fact that patients qualified for RA are often “non-option” patients (due to their comorbidity, they aren’t candidates for CABG in the “every day” practice condition) should be mentioned in the introduction
Please provide a brief discussion of RA as a treatment method for readers who are not interventional cardiologists
The abbreviation of drug-eluting stents seems to be incorrect - please correct it (line 53)
Significant improvement in the percutaneous treatment of ischemic heart disease was achieved after the introduction of second-generation DES - this should be clarified by the Authors
Materials and Methods
Study design section
Please focus on the issues of the follow-up organization (first control visit after 1.5 years ??) - phone calls + central medical databases ?? or scheduled visits in centers performing procedures- clarify it
Clarify indications for RA-PCI – clinical ad hoc decision or rather heart team decision?
Did the patients sign the written consent? If it was not taken for what reason?- (retrospective nature of the study?)
Procedure
Please add a short comment on vascular access point selection
Line 67 “Two patients who did not use RA due to patients’ condition were dropped out from the registry” I don’t understand the meaning of this sentence. Did the patients had unsuccessful RA? or they were initially qualified for the procedure but did not receive it – why? – clarify it
Line 80 please add references
Line 80-84 These two sentences should not appear in the procedure description section
Study outcome
Please, improve your English language and style, some parts of this paragraph are hard to read
Line 88 please expand the RR abbreviation
Results
Table 1.
a) Only 11 patients with a family history of CAD?? Are these data correct?
b) Are there any detailed data for the P2Y12 Inhibitor type, if so, please provide them
c) Any data concerning VKA/NOAC ??
Tabel 2.- the abbreviation LAD (LM) is confusing correct it
It would be interesting from a practical point to know how many procedures were performed by 6 F /7 F access and in how many procedures operators used more than 1 bur size – can you provide this data?
Discussion
Please, improve your English language and style some parts of this paragraph are hard to read
“Technical success rate and the procedural success rate was acceptable, even majority of cases were challenging and risky because of heavily calcified and diffuse coronary lesions”; -in general, I agree that RA is a challenging procedure especially in ACS condition (majority of cases in this registry) however it would be nice to objectify this difficulty, e.g. by expressing the complexity of changes in the syntax scale, percent of a patient initially qualified to CABG or percent of a patient initially pretreated with unsuccessful balloon predilatation
94.0% of all patients received DES during PCI-RA procedure – this is standard treatment - please briefly discuss the cases that did not receive DES - what kind of patients it was?
Line 225- add references.
Line 235-250- there's no need to repeat so many data contained in table 5- they should be “discussed” rather than “provided”.
Big difference in all-cause death rate is mentioned in table 5 studies, did the Authors found a potential reason?
-In the introduction section the authors mention the concern of procedure-related complications and high incidence of adverse event rates. This topic is missing in the discussion section I would like to ask authors to raise this issue in the discussion and refer to the obtained results.
-Relatively high prevalence of OCT/IVUS guided PCI in this “real-life” registry, have Authors seen any benefits of intravascular guidance procedures or can refer the data to other registers
Author Response
We thank the Reviewers for their time and input, and their precious comments.
Reviewers' comments:
Reviewer #1:
Introduction :
I would suggest adding a short comment regarding other methods of calcified lesion modification (NC balloon; cutting/scoring balloon; orbital atherectomy; Shockwave lithotripsy).
Answer : We really appreciate your valuable comments. As you suggested, we revised the manuscript by adding a short comment regarding other methods of calcified lesion modification.
Original text : Line 48
With this current changes, the importance of rotational atherectomy (RA) is emerging.[2]
Modified text : Line 48
With this current changes, the importance of methods for modifying calcified coronary lesions are emerging (such as, non-compliant balloon, cutting/scoring balloon, orbital atherectomy, shockwave lithotripsy and rotational atherectomy [RA]).[2-5]
Original text : Line 287
- Barbato, E.; Carrié, D.; Dardas, P.; Fajadet, J.; Gaul, G.; Haude, M.; Khashaba, A.; Koch, K.; Meyer-Gessner, M.; Palazuelos, J.; et al. European expert consensus on rotational atherectomy. EuroIntervention : journal of EuroPCR in collaboration with the Working Group on Interventional Cardiology of the European Society of Cardiology 2015, 11, 30-36, doi:10.4244/eijv11i1a6.
Modified text :
- Hill, J.M.; Kereiakes, D.J.; Shlofmitz, R.A.; Klein, A.J.; Riley, R.F.; Price, M.J.; Herrmann, H.C.; Bachinsky, W.; Waksman, R.; Stone, G.W. Intravascular Lithotripsy for Treatment of Severely Calcified Coronary Artery Disease. J Am Coll Cardiol 2020, 76, 2635-2646, doi:10.1016/j.jacc.2020.09.603.
- Barbato, E.; Carrié, D.; Dardas, P.; Fajadet, J.; Gaul, G.; Haude, M.; Khashaba, A.; Koch, K.; Meyer-Gessner, M.; Palazuelos, J.; et al. European expert consensus on rotational atherectomy. EuroIntervention 2015, 11, 30-36, doi:10.4244/eijv11i1a6.
- Lee, M.S.; Gordin, J.S.; Stone, G.W.; Sharma, S.K.; Saito, S.; Mahmud, E.; Chambers, J.; Généreux, P.; Shlofmitz, R. Orbital and rotational atherectomy during percutaneous coronary intervention for coronary artery calcification. Catheter Cardiovasc Interv 2018, 92, 61-67, doi:10.1002/ccd.27339.
- De Maria, G.L.; Scarsini, R.; Banning, A.P. Management of Calcific Coronary Artery Lesions: Is it Time to Change Our Interventional Therapeutic Approach? JACC Cardiovasc Interv 2019, 12, 1465-1478, doi:10.1016/j.jcin.2019.03.038.
Also fact that patients qualified for RA are often “non-option” patients (due to their comorbidity, they aren’t candidates for CABG in the “every day” practice condition) should be mentioned in the introduction
Answer : With response to the reviewer’s valuable comments, we mentioned about vulnerable conditions of patients receiving RA procedure.
Original text : Line 50
The clinical outcome of RA have been improved over time, however, operators still concern procedure-related complications and high incidence of adverse event rates.[5-9]
Modified text : Line 53
However, in our daily practice, patients qualified for RA are often “non-option” patients due to their comorbidity, operators should concern about procedure-related complications and high incidence of adverse event rates.[8-12]
Please provide a brief discussion of RA as a treatment method for readers who are not interventional cardiologists
Answer : I totally agree with your opinion. We revised the manuscript with brief discussion of RA
Original text : Line 49
RA is an useful device for plaque modification of severe calcific coronary lesions to facilitate balloon and stent deliver.
Modified text : Line 50
RA system is composed of a high-speed rotating diamond-coated burr that modify calcified coronary lesions. Therefore, it plays as an useful device for plaque modification of those lesions to facilitate balloon and stent deliver.
The abbreviation of drug-eluting stents seems to be incorrect - please correct it (line 53)
Answer : Thank you for your comment. We corrected it to ‘DES’.
Original text : Line 52
With widespread use of drug-eluting stents (DESs)
Modified text : Line 56
With widespread use of second-generation drug-eluting stents (DES)
Significant improvement in the percutaneous treatment of ischemic heart disease was achieved after the introduction of second-generation DES - this should be clarified by the Authors.
Answer : I totally agree with reviewer’s comment. Besides, one of the strong point of our registry is well-representing the clinical outcomes of RA in the second-generation DES era. So, I changed ‘DES’ to ‘second-generation DES’ to emphasize the issue that reviewer pointed out.
Original text : Line 52
With widespread use of drug-eluting stents
Modified text : Line 56
With widespread use of second-generation drug-eluting stents
Materials and Methods
Study design section
Please focus on the issues of the follow-up organization (first control visit after 1.5 years ??) - phone calls + central medical databases ?? or scheduled visits in centers performing procedures- clarify it.
Answer : Thank you for your comments. Because we collected data retrospectively, we clarified it in the original text.
Original text : Line 63
Follow-up data were collected retrospectively, based on medical records and on physician or patient interview.
Clarify indications for RA-PCI – clinical ad hoc decision or rather heart team decision?
Answer : RA-PCI was performed by the operator's clinical ad hoc decision. So, we revised the manuscript to clarify this point as reviewer’s recommend.
Original text : Line 73
The treatment strategy including decisions of burr size, during procedure was at the discretion of the attending cardiologists with careful consideration of clinical risk factors, anatomical complexity, patients’ conditions.
Modified text : Line 80
The treatment strategy including decisions of performing/timing RA, burr size, and selection of vascular access was at the discretion of the attending cardiologists with careful consideration of clinical risk factors, anatomical complexity, patients’ conditions.
Did the patients sign the written consent? If it was not taken for what reason?- (retrospective nature of the study?)
Answer : No, they did not. Because this study was conducted retrospectively.
Procedure
Please add a short comment on vascular access point selection
Answer : With response of your comments, we mentioned about vascular access which also was at the discretion of operators.
Original text : Line 73
The treatment strategy including decisions of burr size, during procedure was at the discretion of the attending cardiologists with careful consideration of clinical risk factors, anatomical complexity, patients’ conditions.
Modified text : Line 80
The treatment strategy including decisions of performing/timing RA, burr size, and selection of vascular access was at the discretion of the attending cardiologists with careful consideration of clinical risk factors, anatomical complexity, patients’ conditions.
Line 67 “Two patients who did not use RA due to patients’ condition were dropped out from the registry” I don’t understand the meaning of this sentence. Did the patients had unsuccessful RA? or they were initially qualified for the procedure but did not receive it – why? – clarify it
Answer : After reviewing angiography, two patients could not use RA were dropped out. One experienced coronary perforation and cardiac tamponade. The other could not pass the wire through the target lesion. I have changed the sentence for better understanding.
Original text : Line 67
Two patients who did not use RA due to patients’ condition were dropped out from the registry.
Modified text : Line 73
After reviewing angiography, two lesions could not be performed RA were dropped out. One experienced coronary perforation and cardiac tamponade. The other could not pass the wire through the target lesion.
Line 80 please add references
Answer : Thank you for your comment. I added references.
Modified text : Line 326
References
- Levine, G.N.; Bates, E.R.; Bittl, J.A.; Brindis, R.G.; Fihn, S.D.; Fleisher, L.A.; Granger, C.B.; Lange, R.A.; Mack, M.J.; Mauri, L.; et al. 2016 ACC/AHA Guideline Focused Update on Duration of Dual Antiplatelet Therapy in Patients With Coronary Artery Disease: A Report of the American College of Cardiology/American Heart Association Task Force on Clinical Practice Guidelines: An Update of the 2011 ACCF/AHA/SCAI Guideline for Percutaneous Coronary Intervention, 2011 ACCF/AHA Guideline for Coronary Artery Bypass Graft Surgery, 2012 ACC/AHA/ACP/AATS/PCNA/SCAI/STS Guideline for the Diagnosis and Management of Patients With Stable Ischemic Heart Disease, 2013 ACCF/AHA Guideline for the Management of ST-Elevation Myocardial Infarction, 2014 AHA/ACC Guideline for the Management of Patients With Non-ST-Elevation Acute Coronary Syndromes, and 2014 ACC/AHA Guideline on Perioperative Cardiovascular Evaluation and Management of Patients Undergoing Noncardiac Surgery. Circulation 2016, 134, e123-155, doi:10.1161/cir.0000000000000404.
- Levine, G.N.; Bates, E.R.; Blankenship, J.C.; Bailey, S.R.; Bittl, J.A.; Cercek, B.; Chambers, C.E.; Ellis, S.G.; Guyton, R.A.; Hollenberg, S.M.; et al. 2011 ACCF/AHA/SCAI guideline for percutaneous coronary intervention: a report of the American College of Cardiology Foundation/American Heart Association Task Force on Practice Guidelines and the Society for Cardiovascular Angiography and Interventions. Catheter Cardiovasc Interv 2013, 82, E266-355, doi:10.1002/ccd.23390.
Line 80-84 These two sentences should not appear in the procedure description section
Answer : I appreciated your comment. I deleted those two sentences in the procedure section.
Modified text : Line 69
2.1. Study Design and Population
Follow-up data were collected retrospectively, based on medical records and on physi-cian or patient interview. This study was approved by the local ethics committee of each hospital.
Study outcome
Please, improve your English language and style, some parts of this paragraph are hard to read
Line 88 please expand the RR abbreviation
Answer : With response to the reviewer’s valuable comments, we spell out RR in the manuscript.
Original text : Line 88
death, any MI, target-vessel spontaneous MI, any RR,
Modified text : Line 93
death, any MI, target-vessel spontaneous MI, any repeat revascularization (RR),
Results
Table 1.
- a) Only 11 patients with a family history of CAD?? Are these data correct?
Answer : I appreciate reviewer’s valuable comment. We rechecked almost half of patients by medical record review. We considered non-accessible states as non-existent. As reviewer pointed out, these data are incorrect. I think it may be a hereditary limitations of retrospective study because in our daily practice, because physicians do not always check the family history of CAD in Korea. We thought that it is not appropriate to show these data and we could omit. To resolve this problem, we respectfully request comments from reviewers.
- b) Are there any detailed data for the P2Y12 Inhibitor type, if so, please provide them
Answer : I am afraid that I should tell you we have no data of type of P2Y12 inhibitors
- c) Any data concerning VKA/NOAC ??
Answer : Thank you for your comment. We collected data of NOAC which was prescribed in 17 (2.9%) patients. We added this in Table 1.
Original text : Line 146
Table 1. Baseline Characteristics of the Patients.
|
Characteristics |
Mean ± SD, n (%) or |
|
Age (years) |
71.4 ± 10.1 |
|
Male |
323 (59.8) |
|
BMI (kg/m2) |
24.2 ± 3.9 |
|
Diabetes mellitus |
305 (56.5) |
|
Insulin use |
72 (13.3) |
|
Hypertension |
415 (76.9) |
|
Hyperlipidemia |
235 (43.5) |
|
Current smoker |
103 (19.1) |
|
Family history of CAD |
11 (2.0) |
|
Clinical presentation |
|
|
Silent ischemia |
37 (6.9) |
|
Stable angina |
176 (32.6) |
|
Unstable angina |
175 (32.4) |
|
NSTEMI |
133 (24.6) |
|
STEMI |
19 (3.5) |
|
Presented as ACS |
327 (60.6) |
|
Chronic kidney disease |
96 (17.8) |
|
Prior MI |
66 (12.2) |
|
Prior PCI |
138 (25.6) |
|
Prior CABG |
24 (4.4) |
|
Peripheral vascular disease |
39 (7.2) |
|
History of heart failure |
77 (14.3) |
|
Atrial fibrillation |
49 (9.1) |
|
LV EF, % |
53.0 ± 13.4 |
|
Medication at discharge |
|
|
Aspirin |
529 (98.0) |
|
P2Y12 Inhibitor |
530 (98.1) |
|
DAPT |
518 (95.9) |
|
B-blocker |
381 (70.6) |
|
ACEI/ARB |
341 (63.1) |
|
Statin |
499 (92.4) |
Values are presented as n (%) or mean ± SD. Abbreviations: SD, standard deviation; IQR, interquartile range; BMI, body mass index; CAD, coronary artery disease; NSTEMI, non-ST elevation myocardial infarction; STEMI, ST elevation myocardial infarction; ACS, acute coronary syndrome; MI, myocardial infarction; PCI, percutaneous coronary intervention; CABG, coronary artery bypass grafting; LV EF, left ventricle ejection fraction; DAPT, dual anti-platelet therapy; ACEI/ARB, angiotensin-converting-enzyme inhibitor/angiotensin II receptor blocker.
Modified text : Line 149
Table 1. Baseline Characteristics of the Patients.
|
Characteristics |
Mean ± SD, n (%) or |
|
Age (years) |
71.4 ± 0.4 |
|
Male |
323 (59.8) |
|
BMI (kg/m2) |
24.2 ± 3.9 |
|
Diabetes mellitus |
305 (56.5) |
|
Insulin use |
72 (13.3) |
|
Hypertension |
415 (76.9) |
|
Hyperlipidemia |
235 (43.5) |
|
Current smoker |
103 (19.1) |
|
Clinical presentation |
|
|
Silent ischemia |
37 (6.9) |
|
Stable angina |
176 (32.6) |
|
Unstable angina |
175 (32.4) |
|
NSTEMI |
133 (24.6) |
|
STEMI |
19 (3.5) |
|
Presented as ACS |
327 (60.6) |
|
Chronic kidney disease |
96 (17.8) |
|
Prior MI |
66 (12.2) |
|
Prior PCI |
138 (25.6) |
|
Prior CABG |
24 (4.4) |
|
Peripheral vascular disease |
39 (7.2) |
|
History of heart failure |
77 (14.3) |
|
Atrial fibrillation |
49 (9.1) |
|
LV EF, % |
53.0 ± 13.4 |
|
Medication at discharge |
|
|
Aspirin |
529 (98.0) |
|
P2Y12 Inhibitor |
530 (98.1) |
|
DAPT |
518 (95.9) |
|
B-blocker |
381 (70.6) |
|
ACEI/ARB |
341 (63.1) |
|
Statin |
499 (92.4) |
|
NOAC |
17 (2.9) |
Values are presented as n (%) or mean ± SD. Abbreviations: SD, standard deviation; IQR, interquartile range; BMI, body mass index; CAD, coronary artery disease; NSTEMI, non-ST elevation myocardial infarction; STEMI, ST elevation myocardial infarction; ACS, acute coronary syndrome; MI, myocardial infarction; PCI, percutaneous coronary intervention; CABG, coronary artery bypass grafting; LV EF, left ventricle ejection fraction; DAPT, dual anti-platelet therapy; ACEI/ARB, angiotensin-converting-enzyme inhibitor/angiotensin II receptor blocker; NOAC, non-vitamin K oral anticoagulant.
Tabel 2.- the abbreviation LAD (LM) is confusing correct it
Answer : With response of your valuable comment, LM was erased to avoid confusion.
It would be interesting from a practical point to know how many procedures were performed by 6 F /7 F access and in how many procedures operators used more than 1 bur size – can you provide this data?
Answer : It is a very excellent point that I did not realize. We already have data of access size (5-8 F) and number of used burr. We provided those data in Table 2.
Original text : Line 159
Table 2. Lesion and Procedural Characteristics.
|
Characteristics |
Mean ± SD, n (%) or |
|
ACC/AHA classification |
|
|
Type B2/C |
59 (10.1)/480 (82.4) |
|
Target vessel |
|
|
LAD (LM) |
384 (65.9) |
|
LCX |
59 (10.1) |
|
RCA |
140 (24.0) |
|
LM disease |
82 (14.1) |
|
Multivessel disease |
467 (80.1) |
|
Femoral approach |
323 (55.4) |
|
Treatment type |
|
|
POBA |
6 (1.0) |
|
DEB |
17 (2.9) |
|
Stent implantation |
548 (94.0) |
|
Failure |
12 (2.0) |
|
IVUS or OCT |
268 (46.0) |
|
Maximum Burr size, mm |
|
|
1.25 |
157 (26.9) |
|
1.5 |
308 (52.9) |
|
1.75 |
107 (18.4) |
|
2.0 |
10 (1.7) |
|
2.5 |
1 (0.2) |
|
Total number of stents |
2.38 ± 1.18 |
|
Number of stents (target-vessel) |
1.66 ± 0.68 |
|
Mean stent diameter, mm (target-vessel) |
2.99 ± 0.38 |
|
Total stent length, mm |
69.02 ± 35.66 |
|
Stent length, mm (target-vessel) |
49.48 ± 21.13 |
Abbreviations: SD, standard deviation; ACC/AHA, American College of Cardiology/ American Heart Association; LAD, left anterior descending; LM, left main; LCX, left circumflex; RCA, right coronary artery; POBA, plain old balloon angioplasty; DEB, drug-eluting balloon; IVUS, intravascular ultrasound; OCT, optical coherence tomography.
Modified text : Line 163
Table 2. Lesion and Procedural Characteristics.
|
Characteristics |
Mean ± SD, n (%) or |
|
ACC/AHA classification |
|
|
Type B2/C |
59 (10.1)/480 (82.4) |
|
Target vessel |
|
|
LAD |
384 (65.9) |
|
LCX |
59 (10.1) |
|
RCA |
140 (24.0) |
|
LM disease |
82 (14.1) |
|
Multivessel disease |
467 (80.1) |
|
Femoral approach |
323 (55.4) |
|
Treatment type |
|
|
POBA |
6 (1.0) |
|
DEB |
17 (2.9) |
|
Stent implantation |
548 (94.0) |
|
Failure |
12 (2.0) |
|
IVUS or OCT |
268 (46.0) |
|
Escalation of Burr |
91 (15.6) |
|
Maximum Burr size, mm |
|
|
1.25 |
157 (26.9) |
|
1.5 |
308 (52.9) |
|
1.75 |
107 (18.4) |
|
2.0 |
10 (1.7) |
|
2.5 |
1 (0.2) |
|
Access size |
|
|
5 F |
5 (0.9) |
|
6 F |
215 (36.9) |
|
7 F |
326 (55.9) |
|
8 F |
37 (6.3) |
|
Total number of stents |
2.38 ± 1.18 |
|
Number of stents (target-vessel) |
1.66 ± 0.68 |
|
Mean stent diameter, mm (target-vessel) |
2.99 ± 0.38 |
|
Total stent length, mm |
69.02 ± 35.66 |
|
Stent length, mm (target-vessel) |
49.48 ± 21.13 |
Abbreviations: SD, standard deviation; ACC/AHA, American College of Cardiology/ American Heart Association; LAD, left anterior descending; LCX, left circumflex; RCA, right coronary artery; POBA, plain old balloon angioplasty; DEB, drug-eluting balloon; IVUS, intravascular ultrasound; OCT, optical coherence tomography.
Discussion
Please, improve your English language and style some parts of this paragraph are hard to read
“Technical success rate and the procedural success rate was acceptable, even majority of cases were challenging and risky because of heavily calcified and diffuse coronary lesions”; -in general, I agree that RA is a challenging procedure especially in ACS condition (majority of cases in this registry) however it would be nice to objectify this difficulty, e.g. by expressing the complexity of changes in the syntax scale, percent of a patient initially qualified to CABG or percent of a patient initially pretreated with unsuccessful balloon predilatation.
Answer : I appreciated the reviewer’s comments which is really helpful for refining and improving the quality of article. We reviewed angiographic and procedural findings of all cases in core lab. All cases except for a few chronic total occlusion or in-stent restenosis cases were calcified and diffuse disease, which represented by 82.4% of lesions were classified as type C (ACC/AHA classificiation). In 38% of cases, RA was done due to unsuccessful 2.0 mm sized or bigger balloon predilatation. In the others, RA was done promptly by operators decision, which was based on angiographic findings, calcification arc revealed by intravascular images, friction of smaller balloons or wires, and so on.
Original text : Line 214
Technical success rate and procedural success rate was acceptable, even majority of cases were challenging and risky because of heavily calcified and diffuse coronary le-sions; 2) RA during PCI was safe and efficient for revascularization regarding to in-hospital MACCEs and procedural complications as well as long-term clinical out-comes;
Modified text : Line 216
1) Technical and procedural success rate of RA was acceptable in heavily calcified and diffuse coronary lesions; 2) RA during PCI was safe and efficient for revascularization regarding to in-hospital MACCEs and procedural complications as well as procedural outcomes (procedure time, contrast amount, or radiation dose); …
… (Line 230)
All cases except for a few chronic total occlusion or in-stent restenosis cases were calcified and diffuse disease, which represented by 82.4% of lesions were classified as type C (ACC/AHA classificiation). In 38% of cases, RA was done due to unsuccessful 2.0 mm sized or bigger balloon predilatation. In the others, RA was done promptly by opera-tors decision, which was based on angiographic findings, calcification arc revealed by intravascular images, friction of smaller balloons or wires, and so on. Although, we provided data of acceptable efficacy outcomes regarding success rate and procedural out-comes. We achieved final TIMI3 in 96.4% of all cases, which is similar results in the previous studies.[1,12,17-19] The absolute values of procedure time and contrast amount were small compared to the previous study.[18]
94.0% of all patients received DES during PCI-RA procedure – this is standard treatment - please briefly discuss the cases that did not receive DES - what kind of patients it was?
Answer : Thank you for your comments. We thought it should be mentioned in the ‘result’ section. Previously, I mentioned about POBA & DEB cases. With response to your comment, I added the information of those with failed cases.
Original text : Line 165
Stent implantation was performed in 94.0% of patients and plain old balloon angioplasty (POBA) or drug-eluting balloon (DEB) accounted for 3.9%.
Modified text : Line 169
Stent implantation was performed in 548 (94.0%) patients, plain old balloon angioplasty (POBA) in 6 (1.0%) and drug-eluting balloon (DEB) in 16 (2.7%). The others (11 [1.9%] patients) were received nothing.
Line 225- add references.
Answer : We added references.
Original text : Line 223
As described previously, DAPT and statin were prescribed above 90% of study popula-tion, which reflected that these patients were treated as current guidelines.
Modified text : Line 228
As described previously, DAPT and statin were prescribed above 90% of study popula-tion, which would reflect that these patients were treated as current guidelines.[14,17,18]
Line 235-250- there's no need to repeat so many data contained in table 5- they should be “discussed” rather than “provided”.
Answer : With response to reviewer’s valuable comment, we discussed novelty of our registry compared to other registries.
Original text : Line 235
Okai et al.[[1]] reported clinical outcomes of DES era from J2T Multicenter registry, which included 1090 patients. During 3.8 years (IQR, 1.9-6.1 years) of follow-up, major adverse cardiac event (MACE), the composite all-cause death, acute coronary syndrome (ACS), ST, CVA, or TVR occurred in 45.5%. All-cause death occurred in 24.2%, including 10.9% of cardiac deaths. TVR occurred in 21.4%, CVA in 4.7% and ST in 1.3%. Kawamoto et al.[[14]] also reported from the ROTATE registry, which enrolled 985 pa-tients and included 8.8% of bare-metal stents (BMSs). At 2 years, MACE, the composite of all-cause death, follow-up MI, and TLR occurred in 24.9%, all cause death in 9.5%, follow-up MI in 3.3%, TVR in 19.8%, TLR in 16.6%, CVA in 1.0%, and ST in 1.8%. Cor-tese at al.[[15]] also showed from the ROTALINK I study with 1397 patients in BMS (40.7%) and DES (42.3%) era. During mean 2.4 year (range 0.8-5.7 years) of follow-up, among patients with DES era, MACE, the composite outcome of cardiac death, non-fatal MI, and TLR was observed in 15.1%, death in 6.7%, MI in 0.4%, TLR in 8%, and ST in 1.5% from the DES era. Abdel-Wahab et al.[[9]] showed long-term clinical outcome of RA in DES era from single center. During 1.3 years (range 0.08-7 years) of follow-up MACE was 20.7%. Death occurred in 9.0%, MI in 2.7%, TVR in 11.2%, and TLR in 7.9%.
Modified text : Line 257
Patients received 2nd-generation DES were included in 36.3% (J2T Multicenter reg-istry)[1] and 69.3% (ROTATE registry).[20] In our registry, as mentioned above, we included 99.8% of 2nd-generation DES among patients treat with stent implantation. This homogeneity of implanted stents is the unique and precious strength of our registry. Also, nearly half of cases were performed procedures with intravascular image guidance. We could evaluate the benefit of intravascular guidance procedures with our data.
Big difference in all-cause death rate is mentioned in table 5 studies, did the Authors found a potential reason?
Answer : We found one of possible explanations from baseline clinical demographics, which might be due to large proportion (more than 50%) of acute coronary syndrome (ACS) and diabetes mellitus.
Original text : Line 236
Okai et al. reported clinical outcomes of DES era from J2T Multicenter registry, which included 1090 patients.
Modified text : Line 253
As shown in Table 5, our study showed relatively high mortality which was similar to that of J2T Multicenter registry.[1] We found one of possible explanations from baseline clinical demographics in both of two studies, which might be due to large proportion (more than 50%) of acute coronary syndrome (ACS) and diabetes mellitus.
-In the introduction section the authors mention the concern of procedure-related complications and high incidence of adverse event rates. This topic is missing in the discussion section I would like to ask authors to raise this issue in the discussion and refer to the obtained results.
Answer : Thank for reviewer’s precious comment. We added the discussion of in-hospital MACCEs and procedural complication.
Original text : Line 232
Although, in-hospital MACCEs and procedural complications showed feasible and ac-ceptable results compared to other previous studies,[1,9,14,15] which demonstrated the safety and efficiency of RA.
Modified text : Line 246
However, in-hospital MACCEs was developed in 10.6% of our registry, mainly driven by peri-procedural MI, which showed higher events compared to those of previous studies. We thought plausible explanations as follow; 1) because we included more than 60% of ACS patients; 2) recorded events according to 10 times elevation of cardiac enzyme (CK-MB) with medical records, would be over-estimated. Coronary perfora-tion was occurred in 1.9% of cases, which was consistent with previous studies (0.5-2.3%).[12,19-21]
-Relatively high prevalence of OCT/IVUS guided PCI in this “real-life” registry, have Authors seen any benefits of intravascular guidance procedures or can refer the data to other registers
Answer : We preliminarily analyzed the benefit of intravascular guidance procedures in RA, which showed no statistically significant difference. We are preparing the article with this issue after PS-matching analysis. We added the information in the manuscript.
Original text : Line 250
… In our registry, TVF occurred in 16.0% at 1.5 year,
Modified text : Line 260
Also, nearly half of cases were performed procedures with intravascular image guidance. We could evaluate the benefit of intravascular guidance procedures with our data.
Reviewer 2 Report
This is a multicenter registry study in Korea that evaluated the short- and long-term outcomes of percutaneous coronary intervention with rotational atherectomy (RA). The incidence of major cerebral and cardiovascular events during hospitalization was 10.8%, and the incidence of target vessel failure (TVF) was 16% at a median follow-up of 1.5 years. the incidence of TVF was independently associated with current smoking, chronic kidney disease, history of cerebrovascular attack, low left ventricular ejection fraction, and left main lesions. The authors conclude that the use of RA is safe and worthwhile.
The methodology of this study is standard and the results are consistent. As shown in Table 5, the results of this study are comparable to data from similar registry studies in the past, and do not question the feasibility of RA for coronary lesions with high degree of calcification. On the other hand, the data presented in this study is merely a cloning of studies that have already been reported in the past, and is not novel. The ROTATE registry - an Italian multicenter registry comparing the outcomes of RA in non-STEMI and stable angina - showed that long-term outcomes were inferior in non-STEMI patients while short-term outcomes were similar in both treatment groups (PMID. 27998837). This study, however, showed only two-thirds use of second-generation DES. I would be personally interested to know how this increased proportion affects the long-term outcomes. If possible, why not explore ACS patients a bit more as a substudy?
Minor revisions are listed below.
Page 3, line 112: Please spell out RR.
Page 3, line 121: Please spell out SD.
Page 3, line 130: p-value should be consistent between lowercase and uppercase (in all manuscripts).
Page 3, line 137: Mean age is not linked to Table 1.
Page 9, line 253: Data showing cardiogenic shock is not presented in the patient background.
Table 5: The citation for the Rathore et al. report is not shown (possibly PMID: 20432398).
Figure 2: It is a matter of preference whether the vertical axis of each figure should be consistent or not, but I think it is easier to see if the percentages of the vertical axis are aligned so that they can be compared.
Author Response
We thank the Reviewers for their time and input, and their precious comments.
Reviewer #2:
This is a multicenter registry study in Korea that evaluated the short- and long-term outcomes of percutaneous coronary intervention with rotational atherectomy (RA). The incidence of major cerebral and cardiovascular events during hospitalization was 10.8%, and the incidence of target vessel failure (TVF) was 16% at a median follow-up of 1.5 years. The incidence of TVF was independently associated with current smoking, chronic kidney disease, history of cerebrovascular attack, low left ventricular ejection fraction, and left main lesions. The authors conclude that the use of RA is safe and worthwhile.
The methodology of this study is standard and the results are consistent. As shown in Table 5, the results of this study are comparable to data from similar registry studies in the past, and do not question the feasibility of RA for coronary lesions with high degree of calcification. On the other hand, the data presented in this study is merely a cloning of studies that have already been reported in the past, and is not novel. The ROTATE registry - an Italian multicenter registry comparing the outcomes of RA in non-STEMI and stable angina - showed that long-term outcomes were inferior in non-STEMI patients while short-term outcomes were similar in both treatment groups (PMID. 27998837). This study, however, showed only two-thirds use of second-generation DES. I would be personally interested to know how this increased proportion affects the long-term outcomes. If possible, why not explore ACS patients a bit more as a substudy?
Answer : We really appreciate the reviewer’s valuable comments. The K-M curve revealed that patients with ACS showed the poorer clinical outcomes regarding TVF compared with SA (log-rank p=0.03).
Minor revisions are listed below.
Page 3, line 112: Please spell out RR.
Answer : With response to the reviewer’s valuable comments, we spell out RR in the manuscript.
Original text : Line 88
death, any MI, target-vessel spontaneous MI, any RR,
Modified text : Line 93
death, any MI, target-vessel spontaneous MI, any repeat revascularization (RR),
Page 3, line 121: Please spell out SD.
Answer : With response to the reviewer’s valuable comments, we spell out SD in the manuscript.
Original text : Line 121
Continuous variables were presented as mean ± SD and compared
Modified text : Line 124
Continuous variables were presented as mean ± standard deviation (SD) and compared
Page 3, line 130: p-value should be consistent between lowercase and uppercase (in all manuscripts).
Answer : The ‘p-value’ has been consistent in lowercase in all manuscript.
Page 3, line 137: Mean age is not linked to Table 1.
Answer : The error was corrected in Table 1.
Page 9, line 253: Data showing cardiogenic shock is not presented in the patient background.
Answer : With response to the reviewer’s valuable comments, we rechecked the raw data and concluded that low systolic blood pressure cannot be considered as cardiogenic shock. So, we erased that phrase
Original text : Line 253
including ST-elevation MI or cardiogenic shock, and the longest follow-up period was 10 year.
Modified text : Line 263
ACS patients including ST-elevation MI, and the long-term follow-up period.
Table 5: The citation for the Rathore et al. report is not shown (possibly PMID: 20432398).
Answer : The citation was corrected in Table 5.
Figure 2: It is a matter of preference whether the vertical axis of each figure should be consistent or not, but I think it is easier to see if the percentages of the vertical axis are aligned so that they can be compared.
Answer : With response of your comment, all vertical axes have been changed to be uniform.
Round 2
Reviewer 1 Report
Significant improvement in the quality of the manuscript - congratulations to the authors. In my opinion, minor English language and style corrections could increase the value of the paper.
Author Response
Significant improvement in the quality of the manuscript - congratulations to the authors. In my opinion, minor English language and style corrections could increase the value of the paper.
Answer : It was an honor to receive the guidance of this reviewer. Thank you very much. I will try my best to improve English language and style.
Reviewer 2 Report
I am pleased that the first revision has been satisfactorily handled with sincerity by the authors.
I am very sorry that my points were not properly explained. I am concerned about the academic necessity for the authors to report to the world on this study as it stands, since it does not have an absolutely large number of cases compared to previous studies, nor does it use a new research approach. Simply put, the authors' registry has a higher percentage of ACS and a higher use of 2nd-generation DES compared to previous studies, which I think are strengths that differentiate them.In the discussion section, the authors indicate that the high mortality rate shown in Table 5 (another question, where should the 27.1% of all deaths be referenced?). Since the rate of DM is generally not significantly different from previous studies, I believe that the ACS rate contributes to the high mortality rate.In their response to the review, the authors indicated that the outcomes in ACS were inferior to those in SA, but I think this is a point of shared interest and should be properly disclosed and emphasized. To be specific, why don't you present the KM curve in Figure 2 grouped by ACS and SA as supplementary material?
Author Response
I am pleased that the first revision has been satisfactorily handled with sincerity by the authors.
Answer : Thank you for your kind and attentive advice.
I am very sorry that my points were not properly explained. I am concerned about the academic necessity for the authors to report to the world on this study as it stands, since it does not have an absolutely large number of cases compared to previous studies, nor does it use a new research approach. Simply put, the authors' registry has a higher percentage of ACS and a higher use of 2nd-generation DES compared to previous studies, which I think are strengths that differentiate them.
Answer : I totally agree with the reviewer’s comment. Higher proportion of ACS patients and 2nd-generation DES are strengths of our registry.
In the discussion section, the authors indicate that the high mortality rate shown in Table 5 (another question, where should the 27.1% of all deaths be referenced?).
Answer : We decided to change contents of Table 5 to event rates at 18 months, which was median of follow-up duration. Because heterogeneity of follow-up duration (0-9.98 years, interquartile range 0.7-2.9 years), previously represented event rates had inherent bias by this issue
Table 5. Comparison with previous studies of Rotational Atherectomy including Technical success and Follow-up Clinical outcomes.
|
|
All study population |
Era |
Technical success, % |
Follow-up duration |
All-cause death, % |
TVR, % |
TLR, % |
ST, % |
CVA, % |
|
Okai et al.[1] |
1090 |
DES |
96.2 |
3.8 (IQR 1.9-6.1) years |
24.2 |
21.4 |
16.2 |
1.3 |
4.7 |
|
Kawamoto et al.[20] |
985 |
8.8% BMS |
99.1 |
2 years |
9.5 |
19.8 |
16.6 |
1.8 |
1.0 |
|
Cortese et al.[21] |
1397 |
17% POBA |
99.3 |
2.4 (0.8-5.7) years |
6.4 |
- |
11.7 |
1.4 |
- |
|
Rathore et al.[19] |
516 |
24.2% BMS |
99.4 |
6-9 months |
- |
- |
14.4 |
0.8* |
- |
|
Abdel-Wahab et al.[12] |
205 |
DES |
98.0 |
1.3 (0.08-7) years |
9.0 |
11.2 |
7.9 |
1.0 |
- |
|
Our study** |
540 |
DES |
96.4 |
1.5 (IQR 0.7-2.9) years |
8.4 |
9.8 |
8.2 |
1.2 |
2.0 |
*Includes only late stent thrombosis.
**Event rates at 18 months were estimated on Kaplan–Meier estimates in time-to-first-event analyses.
TVR, target-vessel revascularization; TLR, target-lesion revascularization; ST, stent thrombosis; CVA, cerebrovascular accident; DES, drug-eluting stent; IQR, interquartile range; BMS, bare metal stent; POBA, plain old balloon angioplasty
Since the rate of DM is generally not significantly different from previous studies, I believe that the ACS rate contributes to the high mortality rate. In their response to the review, the authors indicated that the outcomes in ACS were inferior to those in SA, but I think this is a point of shared interest and should be properly disclosed and emphasized. To be specific, why don't you present the KM curve in Figure 2 grouped by ACS and SA as supplementary material?
Answer : With response of reviewer’s comments, we presented the KM curve in Supplement Figure 1 grouped by ACS and SA. Thank you for your valuable comments.
In manuscript, we added new content of subgroup analysis.
3.4. Subgroup analysis among patients received PCI successfully according to clinical presentation.
To demonstrate the high mortality rate of our registry (27.1% during entire follow-up period [0-10.1 years]), we performed subgroup analysis of patients who received PCI successfully (n=541). We investigated the long-term clinical outcomes according to clinical presentation (ACS versus stable angina). Kaplan-Meier curves showed poorer clinical outcomes regarding the TVF and mortality in ACS group compared to stable angina group (21.7% vs. 14.5%, log-rank p=0.005; 14.1% vs. 6.8%, log-rank p<0.001, respectively) (Supplement Figure 1). Multivariable cox regression analysis revealed that ACS was an independent predictor of mortality (HR, 1.94; 95% CI, 1.002-3.75; p=0.049) (Supplement Table 3).
In Supplement file, we added it as well as tables.
Supplement Table 1. Clinical characteristics according to clinical presentation among patients received PCI successfully.
|
|
ACS |
SA |
p value |
|
Age (years) |
72.4 ± 9.6 |
70.7 ± 9.8 |
0.049 |
|
Male |
188 (57.7) |
141 (65.6) |
0.07 |
|
BMI (kg/m2) |
24.0 ± 3.8 |
24.6 ± 4.0 |
0.07 |
|
Diabetes melltus |
183 (56.1) |
120 (55.8) |
0.94 |
|
Insulin use |
41 (12.6) |
31 (14.4) |
0.54 |
|
Hypertension |
252 (77.3) |
171 (79.5) |
0.54 |
|
Hyperlipidemia |
132 (40.5) |
110 (51.2) |
0.015 |
|
Current smoker |
62 (19.0) |
41 (19.1) |
0.99 |
|
Chronic kidney disease |
61 (18.7) |
39 (18.1) |
0.87 |
|
Prior MI |
40 (12.3) |
26 (12.1) |
0.95 |
|
Prior PCI |
77 (23.6) |
60 (27.9) |
0.26 |
|
Prior CABG |
15 (4.6) |
8 (3.7) |
0.62 |
|
Peripheral vascular disease |
17 (5.2) |
23 (10.7) |
0.017 |
|
History of heart failure |
47 (14.4) |
33 (15.3) |
0.77 |
|
History of CVA |
56 (17.2) |
20 (9.3) |
0.01 |
|
Atrial fibrillation |
31 (9.5) |
18 (8.4) |
0.65 |
|
LV EF, % |
51.4 ± 13.7 |
55.1 ± 12.9 |
0.002 |
Values are presented as n (%) or mean ± SD.
Abbreviations: SD, standard deviation; ACS, acute coronary syndrome; SA, stable angina; BMI, body mass index; MI, myocardial infarction; PCI, percutaneous coronary intervention; CABG, coronary artery by-pass grafting; CVA, cerebrovascular attack; LV EF, left ventricle ejection fraction.
Supplement Table 2. Lesion and procedural characteristics according to clinical presentation among patients received PCI successfully.
|
|
ACS |
SA |
p value |
|
ACC/AHA classification |
|
|
|
|
Type B2/C |
310 (95.1) |
196 (91.2) |
0.07 |
|
Target vessel |
|
|
0.27 |
|
LAD |
229 (70.2) |
137 (63.7) |
|
|
LCX |
26 (8.0) |
19 (8.8) |
|
|
RCA |
71 (21.8) |
59 (27.4) |
|
|
LM disease |
51 (15.6) |
29 (13.5) |
0.49 |
|
Multivessel disease |
275 (84.4) |
167 (77.7) |
0.049 |
|
Femoral approach |
178 (54.6) |
120 (55.8) |
0.78 |
|
IVUS or OCT |
139 (42.6) |
114 (53.0) |
0.018 |
|
Number of stents (target-vessel) |
1.64 ± 0.69 |
1.69 ± 0.66 |
0.40 |
|
Mean stent diameter, mm (target-vessel) |
2.96 ± 0.35 |
3.06 ± 0.43 |
0.004 |
|
Stent length, mm (target-vessel) |
49.42 ± 21.65 |
49.99 ± 20.44 |
0.76 |
Values are presented as n (%) or mean ± SD.
Abbreviations: SD, standard deviation; ACS, acute coronary syndrome; SA, stable angina; ACC/AHA, American College of Cardiology/ American Heart Association; LAD, left anterior descending; LCX, left circumflex; RCA, right coronary artery; LM, left main; IVUS, intravascular ultrasound; OCT, optical coherence tomography.
Supplement Table 3. Multivariable cox regression analysis of independent predictors of all-cause death.
|
|
Multivariate* |
|
|
Variables |
HR (95% CI) |
p |
|
Acute coronary syndrome |
1.94 (1.002-3.75) |
0.049 |
|
Age |
1.04 (1.004-1.08) |
0.03 |
|
Chronic kidney disease |
2.36 (1.24-4.50) |
0.009 |
|
History of cerebrovascular attack |
1.99 (1.03-3.85) |
0.041 |
|
Use of intravascular image |
0.45 (0.23-0.88) |
0.021 |
*adjusted by age, body mass index, current smoker, diabetes mellitus, hyperlipidemia, chronic kidney disease, history of cerebrovascular attack, history of heart failure, presented as acute coronary syndrome, atrial fibrillation, left main disease, multivessel disease, use of intravascular image, and mean stent diameter.
HR, hazard ratio; CI, confidence interval.
Supplement Figure 1. Kaplan-Meier curves for 3-year clinical outcomes according to clinical presentation.
ACS, acute coronary syndrome; SA, stable angina; TV, target-vessel; MI, myocardial infarction; TVR, target vessel revascularization.